# Improving the Mechanical Properties of Damaged Hair Using Low-Molecular Weight Hyaluronate

**DOI:** 10.3390/molecules27227701

**Published:** 2022-11-09

**Authors:** Wenjie Qu, Xueping Guo, Guixin Xu, Songyan Zou, Yuwen Wu, Chenyu Hu, Kuan Chang, Jing Wang

**Affiliations:** 1Bloomage Biotechnology Corp., Ltd., Jinan 250101, China; 2Key Laboratory of Synthetic and Biological Colloids, Ministry of Education, School of Chemical and Material Engineering, Jiangnan University, Wuxi 214122, China

**Keywords:** hair damage, hyaluronate, mechanical properties, tensile properties, hydrogen bond

## Abstract

Chemical treatments of hair such as dyeing, perming and bleaching could cause mechanical damage to the hair, which weakens the hair fibers and makes the hair break more easily. In this work, hyaluronate (HA) with different molecular weight (MW) was investigated for its effects on restoring the mechanical properties of damaged hair. It was found that low-MW HA (average MW~42 k) could significantly improve the mechanical properties, specifically the elastic modulus, of overbleached hair. The fluorescent-labeling experiments verified that the low-MW HA was able to penetrate into the cortex of the hair fiber, while high-MW HA was hindered. Fourier transform infrared spectrometry (FT-IR) results implied the formation of additional intermolecular hydrogen bonds in the HA-treated hair. Thermos gravimetric analysis (TGA) indicated that the HA-treated hair exhibited decreased content of loosely bonded water, and differential scanning calorimetry (DSC) characterizations suggested stronger water bonding inside the HA-treated hair, which could alleviate the weakening effect of loosely bonded water on the hydrogen bond networks within keratin. Therefore, the improved elastic modulus and mechanical strength of the HA-treated hair could be attributed to the enhanced formation of hydrogen bond networks within keratin. This study illustrates the capability of low-MW HA in hair damage repair, implying an enormous potential for other moisturizers to be used in hair care products.

## 1. Introduction

Hair damage is a common phenomenon caused by hair grooming, perming, dyeing, bleaching or treatment by various chemicals. Hair damage makes the hair behave differently from its virgin status, including wetting properties, water retention, combing properties and mechanical properties [1,2,3]. For instance, healthy tresses have obviously greater tensile strength, while damaged tresses tend to break easily during daily grooming. The mechanical properties of hair have attracted more research interest since they are correlated to hair breakage and have intensive academic and practical importance [1,4]. Exploring cosmetic ingredients to repair damaged hair and promote its physical properties, especially the mechanical properties of hair, is of great significance.

At the molecular level, hair damage generally indicates the destruction of keratin configuration and chemical bonds in the hair, such as ionic bonds, disulfide bonds and intermolecular hydrogen bonds [5,6,7,8,9]. This destruction consequently results in a decrease in the mechanical properties of hair [2,3,10]. Many efforts have been devoted to the development of active ingredients to repair hair damage, mostly via the repair of chemical bonds. Malinauskyte et al. [11] found that medium and high-molecular-weight keratin peptides could restore moisture and repair internal chemical bonds, which enhanced the mechanical properties of hair. Song et al. [7] found that polycarboxylic acids could establish bonds between the carboxyl groups in dicarboxylic acids and the amine, sulfhydryl and hydroxyl groups in hair keratin, which increased intermolecular forces and repaired the mechanical properties of hair.

In the past decade, hyaluronate (HA) has been attracting more and more interest due to its excellent water retention capability via its hydrogen bonding network with water [12,13]. Although HA has been widely used in skin-care products, there are few studies on its function and application in hair care. Previous hair research regarding HA is mainly focused on hair growth and loss, but the effect on hair repair, especially mechanical properties, has been rarely investigated [14,15,16,17]. As mentioned above, the hydrogen bond is one of the key chemical bonds in hair keratin, which plays an important role in hair damage and repair [18]. Both hydrogen bonds and water content greatly contribute to the mechanical properties of hair. Therefore, HA could be a promising active ingredient for the repair of hair stretch properties [19,20].

The current paper investigated the effects of HA with different molecular weight on the mechanical properties of hair. To explain the difference in hair repair efficacy, the penetrating behavior of HA with different molecular weight was studied using the fluorescent labeling method. Subsequently, the possible underlying mechanism was investigated using FT-IR, TGA and DSC characterization of HA-treated hair.

## 2. Results

### 2.1. Effect of HA Treatments on Mechanical Properties of Hair

We first checked the ability of different molecular weight (MW) HA to recover the mechanical properties of damaged hair. Tensile strength and Young’s modulus are important parameters to determine hair resistance. Over-bleached Asian hair strands were treated with 0.25% HA solution in a hair-care spray manner ten times, as described in the experimental section. Figure 1a shows the tensile strength of the bleached hair before and after treatment with different MW HA. It was observed that the tensile strength of bleached hair did not show a significant difference after being treated with high-MV and mid-MW HA, but was improved by nearly 16% after being treated with low-MW HA. Moreover, a statistically significant improvement in low-MW HA-treated hair was noticed, demonstrating that low-MW HA could recover the mechanical property of damaged hair. Further analysis of the tensile test results indicated that low-MW HA treatment could significantly increase the Young’s modulus of the bleached hair (Figure 1b). However, all the hair samples exhibited approximately the same elongation rate at break (Figure 1c), suggesting that the improved tensile strength of the low-MW HA-treated hair mainly originated from the increased elastic modulus of hair.

Figure 2 shows the tensile strength and Young’s modulus of the bleached hair treated with hair spray containing varying concentrations of low-MW HA. At low concentrations (<0.25%), low-MW HA improved the tensile strength and Young’s modulus of the bleached hair in a concentration-dependent manner. The results also implied that the improved tensile strength of the HA-treated hair was mainly related to the increased elastic modulus of the hair. In comparison, the peptide-treated hair also present improved Young’s modulus and tensile strength, which is believed to be related to the change in the chemical environment surrounding the hair keratin [21].

### 2.2. Penetration of Labelled HA into Hair Fibers

It was previously reported that different MW peptides could penetrate into the hair fiber with varying penetration efficiency. However, the penetration behavior of polysaccharide molecules with different MW into the hair fibers has been rarely studied [7,11]. In this work, we used fluorescence microscopy as a sensitive technique to investigate the differences in the penetration behavior of different MW HA into hair fibers. Figure 3 shows the cross-sectional photographs of hairs treated with fluorescently labeled HA of high MW and low MW. The images were taken using the green channel, which showed natural autofluorescence of the hair and helped display the positioning of the crosssections in the images. It was observed that the fluorescence intensity of low-MW HA-treated hair was significantly higher than that of high-MW HA-treated hair. Moreover, the low-MW HA penetrated into all parts of the hair, even deep into the cortex, while high-MW HA only entered the outer layers of the fibers. Therefore, the ability of low-MW HA to recover the hair’s mechanical properties might be related to its effective penetration into hair fibers.

### 2.3. FT-IR Characterization of the Low-MW HA-Treated Hair

Figure 4 presents the FT-IR spectra of the bleached hair before and after being treated with low-MW HA. The spectra of both samples showed a broad band at 3272 cm^−1^ attributed to O-H stretching of water or O-H bearing molecules. The intensity of the O-H stretching-related band was significantly increased after the low-MW HA treatment, which could be explained by the additional O-H groups provided by HA molecules. In addition, the amide I band at 1629 cm^−1^ mainly associated with the C=O stretching vibration, and the amide II band at 1517 cm^−1^ associated with the N-H bending vibration and C-N stretching vibration were observed in the bleached hair spectra. The intensity and position of the amide I and II bands are known to be sensitive to the conformation and composition of the keratin. Our results indicated that the amide I/II band intensity ratio increased from 1.036 to 1.092 after HA treatment. Furthermore, the position of the amide II band peak was blue-shifted to 1529 cm^−1^ in HA-treated hair. Since HA treatment would not affect the amino acid or peptide composition of hair, such variation in amide I and II bands might be related to the possible formation of intermolecular H-bonds between HA and keratin.

### 2.4. Effects of HA Treatments on Thermal Properties of Hair

Previous studies suggested that two types of water components, strongly bonded and loosely bonded water, could be differentiated in human hair [22]. The contents of strongly bonded and loosely bonded water were analyzed by thermos gravimetric analysis (TGA), according to the methodology reported by Barba et al. with slight modification. The temperature was increased from 25 °C to 65 °C at a rate of 20 °C/min and maintained at 65 °C for 18 min, followed by an increase up to 180 °C at 20 °C/min and kept at that level for 20 min. The water loss was determined, and the TGA curves were generated. According to the typical TGA curves in Figure 5, the bleached hair and HA-treated hair had similar total water content. However, the loosely bonded water content in the HA-treated hair was slightly lower than that in the untreated bleached hair. The above results indicated that HA treatment affected the existing form of water inside the hair fiber, potentially due to the strong water-binding capacity of HA [23]. The trend of water content is in contrast to the effect of the peptide on hair, which generally increased the loosely bonded water content [24].

DSC was applied to evaluate the thermal behavior of water in hair, which is an effective technique that allows for determining the bonding strength of water in hair fibers [24,25,26]. Figure 6 shows the DSC curves of the bleached hair before and after HA treatment, with the peaks representing different stages of hair degradation. The upward peaks represented endothermic reactions, with the first peaks corresponding to the release of loosely bonded water. The second and third peaks denote endothermic fusion reactions of the keratin polypeptide chain. It was observed that all three peaks in DSC curves were affected by HA treatment. On one hand, the “water evaporation” peak of the HA-treated hair was centered at a higher temperature of 94.8 °C, compared to that of bleached hair (88.3 °C), which is consistent with Margarida’s work, who observed similar peak movement in peptide-treated bleached hair [27]. Moreover, the “water evaporation” peak area of the HA-treated hair, which corresponded to the enthalpy for water vaporization, was obviously larger than that of the bleached hair, suggesting that more energy would be spent during the release of water from HA-treated hair. These results indicated that HA treatment led to stronger water bonding inside the hair fiber. On the other hand, the two peaks corresponding to the melting and decomposition of keratin also shifted to higher temperatures after HA treatment, suggesting that HA treatment also tended to increase the crystallinity of keratin.

## 3. Materials and Methods

### 3.1. Materials

Over-bleached Chinese hair (18 cm × 1 g, free 16 cm) was purchased from Shanghai Canyu Commercial Co., Ltd., (Shanghai, China); sodium hyaluronate with different molecular weight was provided by Bloomage Biotechnology Co., Ltd. (Jinan, China), including high-molecular-weight HA with an average MW of 1460 k (denoted as high-MW HA), middle-molecular-weight HA with an average MW of 370 k (denoted as mid-MW HA) and low-molecular-weight HA with an average MW of 42 k (denoted as Low-MW HA). Other chemicals were purchased from Aladdin (China).

### 3.2. Hair Treatment

Over-bleached Chinese hair strands were washed with 10% (*w/w*) SDS solution and naturally dried at ambient conditions (temperature = 25 ± 2 °C, RH = 50% ± 5%). Different hair strands were treated with 0.3 g hair spray solution containing 0.25% sodium hyaluronate (HA) with different molecular weight. The hair strands treated with deionized water of the same amount were set as the negative control group (bleached hair group). The treated hair strands were stored at ambient conditions for 12 h and then washed with deionized water to remove the free residual HA on the hair surface. The above hair treatment procedure was repeated 10 times before testing the mechanical property of the hair.

### 3.3. Hair Tensile Property Tests

The diameters of hair fibers were measured at the middle section of a single hair fiber by an SN-1200w HD camera (Sannosinu Technology Co., Ltd., Shenzhen, China). For each group of the treated hair strands, 30 single hair fibers with diameters of 90–110 μm were selected from the strands under the micro-camera. The single fiber tensile properties of the hair were then measured by an XS (08) XT-3 fiber strength tester (Shanghai Xusai Instrument Co., Ltd., Shanghai, China). The tensile strength (σ) and strain (ε) of the single hair fiber were calculated through the following formula:σ = F_b_/S(1)
and ε = ΔL/L_0_(2)
where F_b_, S, ΔL and L_0_ are the breaking strength, the cross-sectional area of the fiber, the displacement and the initial length, respectively [10]. Young’s modulus (YM) is a mechanical property, applicable only within Hooke’s law, which is related to tensile strength and strains and provides a measure of material stiffness [9]. The YM of a single hair fiber can be calculated by selecting the Hookean section curve of the derived tensile data and the following formula:E = σ/ε = (F/S)/(ΔL/L)(3)
where that F is the maximum value of strength in the Hookean section. The tensile strength and Young’s modulus of the hair strands were calculated based on the average of 30 hair single fibers. SPSS software was used for statistical analysis, a *t*-test was performed on the data of each group and the control group, and the p value was calculated.

### 3.4. Fluorescent Labeling of HA and Fluorescence Microscopy

To enable the study of HA incorporation into over-bleached hair, fluorescein 5(6)-isothiocyanate (FITC) was linked to HA. Typically, 0.2 g of HA and 0.04 g of FITC were dissolved into 2 mL of 0.05 mol/L NaOH aqueous solution and were then reacted at 95 °C for 45 min. After cooling to room temperature, 18 mL of NaOH-saturated ethanol solution was added and the mixture was centrifuged to obtain crude FITC-labeled HA. The crude FITC-labeled HA was then dispersed into 20 mL of NaCl-saturated ethanol solution and centrifuged to remove the unbound FITC. After alcohol washing 6 times, the precipitation was freeze-dried to obtain purified FITC-labeled HA [28]. Labeled HA was used in the treatment of over-bleached hair as described earlier.

Transversal cuts (15 μm) of hair fiber samples embedded in an epoxy resin were prepared using a microtome (Microtome Leitz, Oberkochen, Germany), and were analyzed by fluorescence microscopy. The cross-section of hair was observed by a fluorescence microscope (Nikon Co., Ltd., Tokyo, Japan). All images were recorded using similar filter, exposure, brightness and gain settings.

### 3.5. Characterization of Hair

Before characterization, the hair strands were kept in a humidity-controlled box (22 °C, 50% RH) for 48 h. Fourier transform infrared spectra of hair fibers were recorded with an FTIR spectrophotometer (Thermo Fisher Scientific, Massachusetts, America) in the spectral range of 1000–4000 cm^−1^ [29]. The moisture content in hair was analyzed by a thermos gravimetric analysis (TGA) instrument (1100SF, Mettler Toledo) in an atmosphere of dry N_2_ where a purge at 30 mL·min^−1^ was employed. For each hair sample, TGA measurements were repeated 5 times [22,30]. The differential scanning calorimetry (DSC) tests were performed on a DSC instrument (NETZSCH Instrument Manufacturing Co., Germany). Typically, 5–10 mg of finely cut hair was rapidly transferred into a DSC capsule with the sample pan cover punched, allowing evaporating water to escape. All the experiments were performed under a constant flow of N_2_. The samples were heated from 30 °C to 300 °C at a rate of 10 °C/min, and the heat flow was recorded [24,25].

## 4. Discussion

Chemical dyeing, including permanent dyeing, greatly disrupts the structural and mechanical properties of hair fibers and causes mechanical damage [31,32]. As a result, the fibers become weak and are more susceptible to breakage with time, which is incompatible with healthy hair. Therefore, the demand for products that improve the fiber qualities of hair is increasing rapidly. One of the simplest ways to assess the integrity and quality of the hair fiber is by measuring its mechanical properties. Indeed, the slightest modification in the chemical composition of hair may greatly alter its mechanical properties [33,34]. Previous works have intensively reported the application of proteins, peptides and amino acids in hair damage repair [35]. However, only a few studies have addressed the effects of polysaccharides on restoring the mechanical properties of damaged hair. In this work, we first reported the potential of low-molecular-weight hyaluronate for hair damage repair and revealed the possible underlying mechanism.

We first examined the effects of HA with different molecular weight on repairing the mechanical properties of over-bleached hair. It was found that only low-MW HA, which showed a higher penetration efficacy, could restore the mechanical properties of damaged hair. Hair fibers have a natural barrier that prevents the penetration of materials into the cortex, and it would be difficult for high-MW compounds to pass through this barrier. The penetration investigation using fluorescence microscopy showed that the low-MW HA was able to penetrate into the cortex, while the high-MW HA only entered the outer layers of the cortex. Therefore, the hair damage repair efficacy of low-MW HA could be related to its ability to penetrate into the hair fibers.

In the previous literature, an equation was proposed to obtain a better quantitative description of the tensile properties of similar α-keratin fibers, which consist of stiff protein fibers and a pliant protein matrix [33,34]. The stretching process of hair was characterized by three stages: (1) a nearly linear region (namely the Hookean region or elastic region) with strains less than 3%, which only involved the change of bond angles without significant structural transformation; (b) a near flat region with little increase in stress (transformation region) due to the α-β transition; (c) the post-transformation region mainly related to the destruction of convent disulphide bonds. In the Hookean region, α-keratin fiber is generally considered a material dominated by hydrogen bonds (H-bonds), and its elastic modulus is highly correlated to the density of effective intermolecular H-bonds. In this work, the mechanical test results in Figure 1 and Figure 2 indicated that low-MW HA treatment could improve the tensile strength of the bleached hair, and such improvement effect was mainly related to the increased elastic modulus rather than the elongation ratio of hair. Therefore, the hair mechanical property-restoring effects of low-MW HA might be associated with the enhanced formation of H-bond networks.

It is well known that the elastic modulus of hair fiber is related to the water content, especially the loosely bonded water content within the fiber [36,37]. Loosely bonded water can disrupt the H-bonding networks, thereby reducing the elastic modulus of the hair fiber. High humidity usually leads to the decreased elastic modulus of hair due to the increased content of the “free water” or loosely bonded water. In this work, HA-treated hair showed a higher elastic modulus than the untreated bleached hair at a fixed test humidity (50%). Since HA has a strong water-binding capacity, it was speculated that HA treatment might affect the existing form of water inside the hair fiber. TGA analysis indicated that the HA-treated hair exhibited a decreased content of loosely bonded water. Moreover, DSC analysis suggested that more energy would be spent during the evaporation and release of water from HA-treated hair, implying stronger water bonding inside the HA-treated hair fibers. Therefore, we proposed that HA treatment led to stronger water bonding inside the hair owing to its high water-binding capacity, thus promoting the formation of hydrogen bond networks within keratins, and thereby leading to the increased elastic modulus of hair.

According to the above discussion, the effect of HA in hair damage repair was tentatively explained by its water-binding ability in the current work. However, according to some previous studies, HA molecules could also directly interact with keratin fibers, which may also contribute to improved mechanical properties of the damaged hair [38]. The results from FT-IR spectra (Figure 4) and DSC analysis (Figure 6) also revealed the structural and crystallinity variation of keratin in the hair fibers, implying the interactions between HA and keratin, especially the possible formation of intermolecular hydrogen bonds between HA and keratin. This study revealed the potential of low-molecular-weight hyaluronate in hair damage repair, and the tentative underlying mechanism was discussed. In the future, more studies will be performed to further investigate the interaction of hyaluronate with hair fiber components.

## 5. Conclusions

Based on the results and discussion presented above, the following conclusions can be made regarding the potential of low-molecular-weight hyaluronate for hair damage repair as well as its working mechanism:(1)Treating damaged hair with HA could significantly improve the mechanical strength, or more specifically, the elastic modulus of hair.(2)Only low MW-HA showed mechanical property-restoring effects on hair, while the effect of HA with high molecular weight was negligible due to its poor penetrating capability.(3)HA treatment led to stronger water bonding inside the hair owing to its high water-binding capacity, therefore promoting the formation of hydrogen bond networks within keratins, and thereby leading to the increased elastic modulus of hair.(4)The possible formation of intermolecular H-bonds between HA and keratin might also contribute to the improved tensile properties of HA-treated hair.

## Figures and Tables

**Figure 1 molecules-27-07701-f001:**
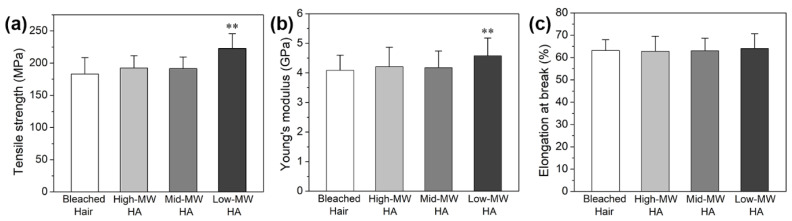
Effects of different MW HA treatments on tensile strength (**a**), Young’s modulus (**b**) and elongation rate (**c**) of bleached hair. (Significance: ** *p* < 0.01 vs. bleached hair. N = 30).

**Figure 2 molecules-27-07701-f002:**
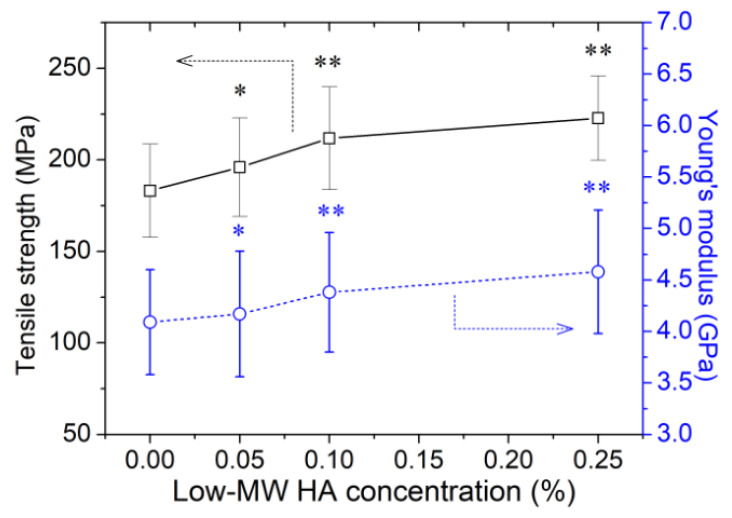
Effect of low-MW HA concentration on the elastic modulus and tensile strength of bleached hair. (Significance: * *p* < 0.05; ** *p* < 0.01 vs. bleached hair. N = 30).

**Figure 3 molecules-27-07701-f003:**
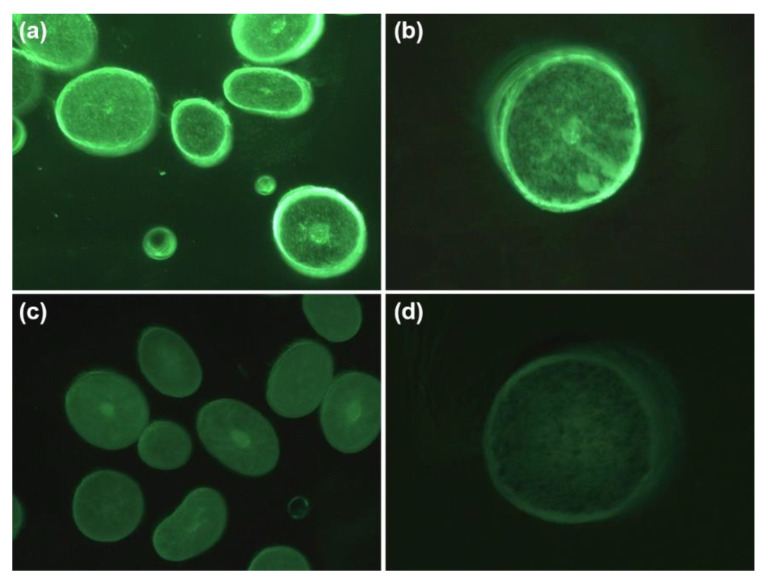
Cross sections of hair treated with fluorescent-labeled low-MW HA (**a**,**b**) and high-MW HA (**c**,**d**).

**Figure 4 molecules-27-07701-f004:**
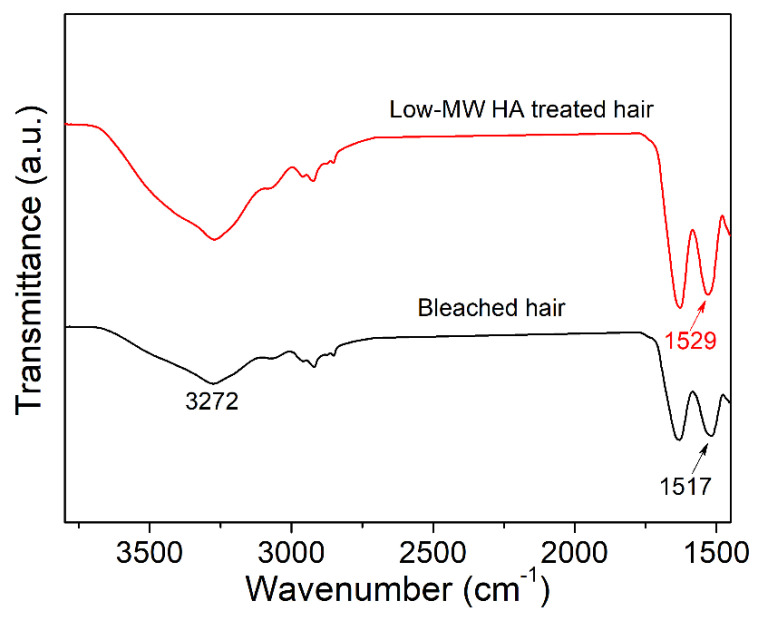
FT-IR spectra of the bleached hair and low-MW HA-treated hair.

**Figure 5 molecules-27-07701-f005:**
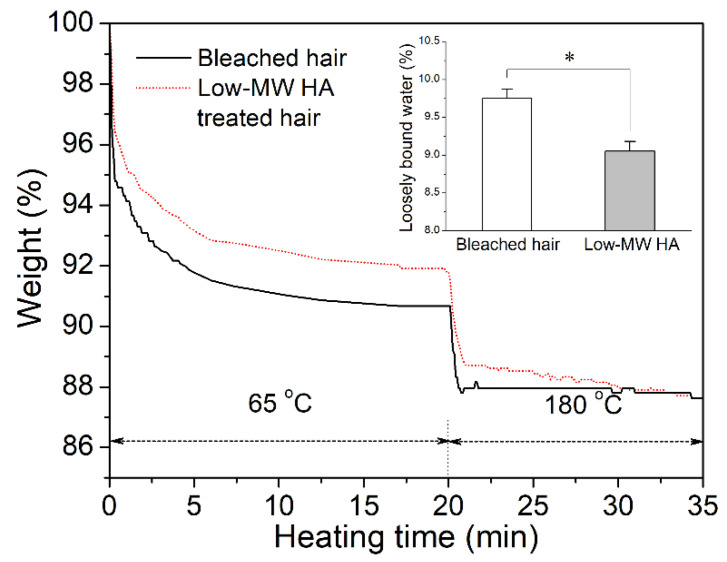
TGA spectra of untreated bleached hair and low-MW HA-treated hair. Inset: average loosely bonded water content in the HA-treated hair and untreated bleached hair. (Significance: * *p* < 0.05 vs. bleached hair. N = 4).

**Figure 6 molecules-27-07701-f006:**
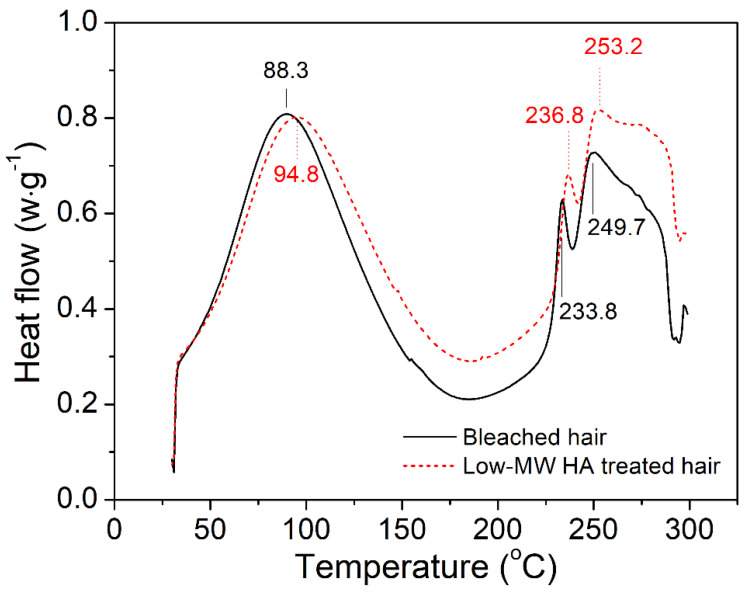
DSC spectra of untreated bleached hair and low-MW HA-treated hair.

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
