# Peer review of "Improving the Mechanical Properties of Damaged Hair Using Low-Molecular Weight Hyaluronate"

_molecules, 2022, doi:10.3390/molecules27227701_

Round 1
Reviewer 1 Report
The highlight for this paper is to investigate the effect of hyaluronate on restoring the mechanical properties of damaged hair. The authors found that low molecular-weight hyaluronate can significantly improve the mechanical properties. These results are interesting and useful.
But as the authors mentioned that ‘Although HA has been widely used in skin-care products, its function and application in hair care has been rarely investigated’. In fact, besides the reference 14, several literatures on this topic can be found, for example, ‘Temporary hair loss after injection of hyaluronic acid filler’, ‘Minoxidil-loaded hyaluronic acid dissolving microneedles to alleviate hair loss in an alopecia’ and ‘in vitro hair growth promoting effect of a noncrosslinked hyaluronic acid in human dermal papilla cells ’ . The authors should include more literatures on this topic than only shown one literature in ref. 14.
Furthermore, for ‘2.3 Hair tensile property tests’, the authors mentioned ‘For each group of the treated hair strands, 30 single hair fibres with diameters 90-110 μm were selected from the strands under the micro-camera’. I am just wondering why only select the hair fibres with diameters between 90 μm and 110 μm? How about using the hair fibres with other diameters (<60 μm or 60~90 μm)?
Therefore,I would like to suggest to publish this article in Molecules after minor revision.
Reviewer 2 Report
Manuscript ID: molecules-2003185
The authors report on the improved mechanical properties of damaged hair using different molecular weight (MW) hyaluronate. The fluorescent labeling examination verified the ability of HA to penetrate into the cortex of the hair fibre. Fourier Transform Infrared Spectrometry (FT-IR) was used to interpret the formation of additional intermolecular hydrogen bonds in the HA-treated hair. The elastic modulus and mechanical strength of the damaged and treated hairs were also evaluated. The thermal properties of those hairs were examined using Thermos Gravimetric Analysis (TGA), and Differential Scanning Calorimetry (DSC) and confirmed the content of loosely bonded water.
It is quite an interesting approach to use molecular analysis to examine bio-macromolecules. I think that the work should be published with regard to a well-thought-out plan and of undoubted interest for practice. However, the presentation of the results needs improvement in accordance with the following comments, which should be addressed prior to publication. Therefore, I am suggesting minor revisions.
1. The title does not reflect the main results. In my opinion, the title should indicate the main results. For instance, “Improving the mechanical properties of damaged hair using low-molecular weight hyaluronate”
2. The authors mentioned the use of keratin peptide to restore the moisture and repair the internal chemical bonds of hair in the INTRODUCTION. In my opinion, the obtained results should be compared with those of keratin peptide or other compounds dedicated to hair treatment. The obtained results, if any, should be discussed in relation to other works.
3. In the Materials and methods, please indicate the MW number of low- and high-MW HA. It is worth to be emphasized.
4. What do the asterisks in Figure 1 mean? Please indicate this in the figure caption, including the statistical method used.
5. Figure 2-difficult to comprehend. The error bars are overlapped. In my opinion, there is no significant difference among the low-MW HA concentrations used.
6. Figure 4-Incompleted phrase. “Low-MW HA” should be “Low-MW HA treated hair”.
7. General remarks;
Line 31 and onward: Please check the reference style of the citation.
Line 41 and 44: Please check the reference style of using author names.
Line 61: The format of this journal, materials and methods should be placed after results and discussion.
